# Cerebellar climbing fibers encode expected reward size

Noga Larry[†]*, Merav Yarkoni[†], Adi Lixenberg, Mati Joshua

Edmond and Lily Safra Center for Brain Sciences, The Hebrew University of Jerusalem, Jerusalem, Israel

**Abstract** Climbing fiber inputs to the cerebellum encode error signals that instruct learning. Recently, evidence has accumulated to suggest that the cerebellum is also involved in the processing of reward. To study how rewarding events are encoded, we recorded the activity of climbing fibers when monkeys were engaged in an eye movement task. At the beginning of each trial, the monkeys were cued to the size of the reward that would be delivered upon successful completion of the trial. Climbing fiber activity increased when the monkeys were presented with a cue indicating a large reward, but not a small reward. Reward size did not modulate activity at reward delivery or during eye movements. Comparison between climbing fiber and simple spike activity indicated different interactions for coding of movement and reward. These results indicate that climbing fibers encode the expected reward size and suggest a general role of the cerebellum in associative learning beyond error correction.

DOI: https://doi.org/10.7554/eLife.46870.001

## Introduction

Computational, anatomical, and functional evidence support the theory that the cerebellar cortex performs error correcting supervised motor learning (*Albus, 1971*; *Gilbert and Thach, 1977*; *Marr, 1969*; *Nguyen-Vu et al., 2013*; *Stone and Lisberger, 1990*; *Suvrathan et al., 2016*). In this framework, motor learning occurs through changes in the computation of Purkinje cells, the sole output cells of the cerebellar cortex. Purkinje cells receive two distinct types of inputs: parallel fiber inputs and climbing fiber inputs. Each type of input leads to a different type of action potential. Parallel fiber inputs modulate the rate of Simple spikes (Sspks), events similar to action potentials in other cell types. Climbing fiber inputs result in complex spikes (Cspks), which are unique prolonged events. Cspks are thought to represent instructive error signals triggered by movement errors. These error signals adjust the Sspk response of the Purkinje cell to parallel fiber input, resulting in improvement in subsequent movements. This hypothesized role of the Cspks in learning was broadened when it was shown that the Cspk rate increases in response to cues that are predictive of undesired successive stimuli (*Ohmae and Medina, 2015*). Thus, the Cspk signal is well-suited for driving associative learning based on motor errors that drive avoidance of aversive stimuli.

Recent research has shown that Cspk rate increases when behavior leads to a desired rewarded outcome or when reward related stimuli are presented (*Heffley et al., 2018*; *Kostadinov et al., 2019*; *Heffley and Hull, 2019*), a marked departure from their established role in error signaling. We aimed to further investigate what is coded by the reward related Cspk increase and whether the reward driven Cspk modulations are linked to simple spike modulations.

We considered three possibilities for the coding of reward by Cspks. The first was that the Cspk reward signal could be directly linked to the physical delivery of reward. For example to reward consumption behavior (such as licking; *Welsh et al., 1995*) or to the signal at reward delivery that behavior was successful (*Heffley et al., 2018*). If so, we would expect reward related modulations of the Cspk rate to be locked to the time of reward delivery. The second possibility was that the Cspks

*For correspondence:
noga.larry@mail.huji.ac.il

[†]These authors contributed equally to this work

**Competing interests:** The authors declare that no competing interests exist.

could encode the predicted reward consequences of arbitrary stimuli, similar to the way in which Cspks encode the prediction of an undesired air-puff (*Ohmae and Medina, 2015*). If this were the case, we would expect a Cspk increase when reward predictive stimuli are presented. Finally, reward could modulate Cspks through the coding of motor errors. In the eye movement system, for instance, Cspks are modulated when the eye velocity does not match the target velocity (i.e. retinal slip; *Stone and Lisberger, 1990*). Reward could influence the representation of the error signal such that similar retinal slips would result in a higher Cspk rate when a greater reward is expected. Thus, if reward acts on error signaling directly, we would expect reward to modulate the Cspk rate at the time of the retinal slip.

To dissociate these alternatives we designed a task that temporally separated reward information, motor behavior and reward delivery (*Joshua and Lisberger, 2012*). We found that climbing fiber activity encoded the expected reward size seconds before the reward delivery. Reward size did not modulate activity at reward delivery. Furthermore, reward expectation did not modulate the Cspk tuning of eye movement parameters. These results suggest the Cspk reward signal encodes changes in the prediction of future reward. During the cue, the modulation in the Cspk and Sspk rates of cells were uncorrelated, in contrast to the negative correlation reported in the context of error correction learning (*Gilbert and Thach, 1977*) or the coding of movement parameters (*Ojakangas and Ebner, 1994*; *Stone and Lisberger, 1990*). This suggests that Cspk modulation of the Sspk rate could be restricted to certain network states. Overall our findings imply that the cerebellum receives signals that could allow it to perform both error and reward-based associative learning, thus going beyond the accepted role of the cerebellum in error correction to suggest a general role in associative learning.

## Results

### Complex spikes encode the size of the expected reward

Monkeys performed a smooth pursuit eye movement task in which we manipulated the expected reward size (*Joshua and Lisberger, 2012*; *Figure 1A*). At the start of each trial, the monkey fixated on a white spot. The spot then changed to one of two colors, indicating whether a large or small reward would be given upon successful completion of the trial. After a variable delay, the colored

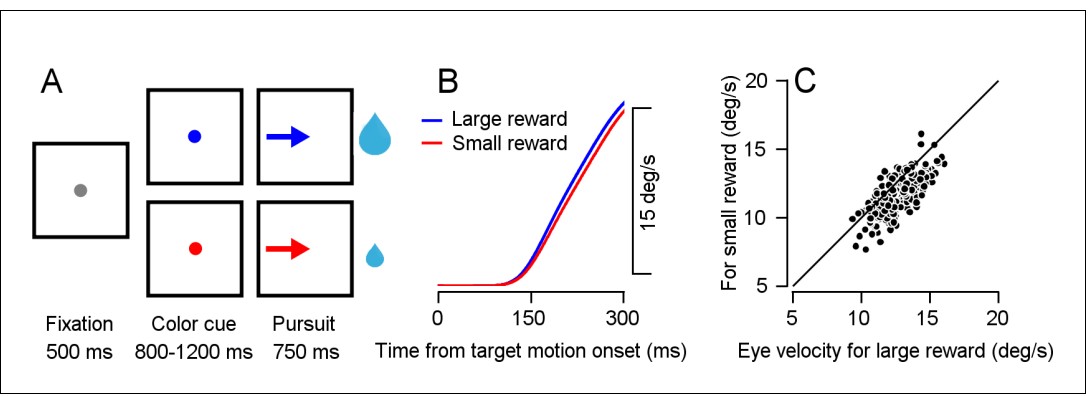

**Figure 1.** Smooth pursuit eye-movement task. (**A**) Eye movement task temporally separates reward expectation, pursuit behavior and reward delivery. (**B**) Traces of average eye speed, in the first 300 ms after target motion onset. Target velocity was 20 °/s. (**C**) Each dot represents the average speed for an individual session 250 ms after target movement onset for the large (horizontal) and small (vertical) reward cue (Signed-rank, p = 8.6*10$^{-24}$, n = 208).
DOI: https://doi.org/10.7554/eLife.46870.002
The following figure supplements are available for figure 1:

**Figure supplement 1.** Monkeys associate reward size with target color.
DOI: https://doi.org/10.7554/eLife.46870.003
**Figure supplement 2.** MRI and examples of extracellular recordings of Cspks.
DOI: https://doi.org/10.7554/eLife.46870.004

target began to move in one of eight directions and the monkey had to accurately track it. At the end of a successful trial, the monkey received either a large or a small reward, as indicated by the color of the cue. To suppress catch-up saccades in the time immediately after the onset of the target movement, the movement of the target was preceded by an instantaneous step in the opposite direction (step-ramp). Thus, when the monkey began tracking, the target was close to the eye position and there was no need for fast corrective eye movements (*Rashbass and Westheimer, 1961*).

The average eye velocity during tracking of the large reward target was faster and more similar to the target velocity than the tracking of the small reward target (*Figure 1B*). This difference was clearly apparent even at the single session level. In most sessions, the average eye velocity of 250 ms following motion onset was larger when the expected reward was large (*Figure 1C*). This behavioral difference and the selection of the larger reward target in an additional choice task (*Figure 1—figure supplement 1*) indicate that the monkeys associated the reward size with the color of the target. During the task, we recorded neural activity from the flocculus complex and neighboring areas (*Figure 1—figure supplement 2*). Our recordings included neurons that responded to eye movements and neurons that did not. Our task design allowed us to separately analyze the Cspk rate following cue presentation, during pursuit, and following reward delivery.

Following cue presentation, we found that many Purkinje cells (40 out of 220) had different Cspk rates in the different reward conditions. Of these, the vast majority (34 cells) transiently increased their Cspk rate when the expected reward was large but not when the expected reward was small

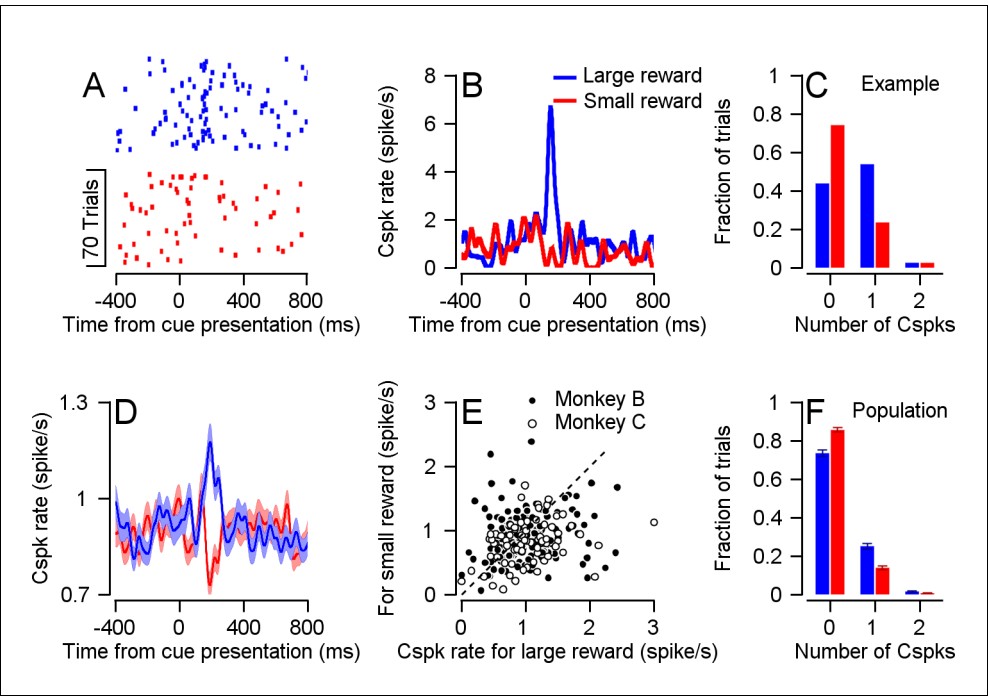

**Figure 2.** Cspk rate differentiates reward conditions during cue presentation. (**A**) Raster plot of an example cell in the two reward conditions, aligned to cue presentation. (**B**) PSTH of the cell in A. (**C**) Histogram of the number of Cspks that occurred in the 100–300 ms time window following cue presentation, in the same example cell. (**D**) Population PSTH. In all figures the error bars represent SEM. (**E**), Each dot represents the average Cspk rate of an individual cell 100–300 ms after the display of the large (horizontal) and small (vertical) reward cue (Signed-rank, Monkey B: p = 0.01, n = 148, Monkey C: p = $3.35*10^{-4}$, n = 72). (**F**) Histogram of the number of Cspks that occurred in the 100–300 ms time window following cue presentation, in the entire population (fraction of trials with 1 Cspks: Signed-rank, p = $5.1*10^{-4}$, n = 40; fraction of trials with two Cspks: Signed-rank, p = 0.03, n = 40).
DOI: https://doi.org/10.7554/eLife.46870.005

The following figure supplement is available for figure 2:

**Figure supplement 1.** Fraction of trials with Cspks following the cue presentation is higher in the large reward condition than in the small reward condition.
DOI: https://doi.org/10.7554/eLife.46870.006

(example in *Figure 2A–C*). This difference was apparent when examining the population average Cspk peri-stimulus time histogram (PSTH). After the color cue appeared, the population average Cspk rate was higher when the expected reward was large, as can be seen by the difference in the PSTHs of the two reward conditions (*Figure 2D*). At the single cell level, most cells had a higher Cspk rate on large reward trials than on small reward trials (*Figure 2E*, most dots lie beneath the identity line). Thus, the Cspk rate was modulated by changes in reward expectation, at times temporally distinct from the behavioral effect on pursuit eye movements and reward delivery. This change of rate reflects mostly an increase in the number of trials with a single Cspk following the cue, and a minor increase in the number of trials with multiple Cspks (*Figure 2C,F* and *Figure 2—figure supplement 1*).

## Complex spikes do not encode reward size at reward delivery

The population Cspk rate was only affected by reward size when information regarding future reward was given, but not during the reward itself. During reward delivery, the PSTHs of the two conditions overlapped (*Figure 3A*), indicating a similar population response for the large and small rewards. When examining the responses of single cells, the Cspk rate was similar in the two reward conditions (*Figure 3B*, most cells fell close to the identity line). To compare the temporal pattern of the reward size encoding at cue and reward delivery, we calculated the difference in PSTHs between the large and small reward conditions (*Figure 3C*). The difference between large and small rewards rose steeply shortly after the color cue appeared. In sharp contrast, following reward delivery, there was only a small rate fluctuation that resembled the fluctuation prior to reward delivery. At the single cell level, there was no correlation between cell encoding of reward size during the cue and during reward delivery. For both the full population and for the subpopulation of neurons significantly coding the reward size at cue, the correlation between cue and reward delivery epochs was not significant (*Figure 3D*). This indicates that Purkinje cells that differentiated reward conditions during the cue did not differentiate between them during delivery.

We ruled out the possibility that differences in licking behavior were responsible for the Cspk rate modulations. The pattern of licking (*Figure 3E,F*) and Cspk rate was completely different. Licking but not spiking increased at reward delivery. Further, after cue onset, licking in both reward conditions decreased whereas the temporal pattern of Cspks was different between reward conditions (*Figure 2D*). In approximately half of the recording sessions, we recorded licking behavior along with our electrophysiological recordings. For the cells that discriminated between reward conditions in these sessions (n = 21), the population PSTH showed a difference between reward conditions both in trials that included a lick immediately following the cue and trials that did not (*Figure 3—figure supplement 1A,B*). We also approximated the contribution of licking to the Cspk rate (*Figure 3—figure supplement 1C,D*). This contribution was negligible and was not different for large and small rewards.

We conducted a similar analysis for saccades and microsaccades. The pattern of saccades and microsaccades also differed from the Cspk pattern (*Figure 3—figure supplement 2A,B*). Saccades but not spiking increased following reward delivery. After cue presentation, fixational saccades were modulated by reward (*Joshua et al., 2015*), but this modulation did not affect the Cspk response to the cue (*Figure 3—figure supplement 2C,D*). The cells that discriminated between the large and small rewards after cue presentation responded similarly in trials with and without saccades. Similar to licking, the approximated contribution of saccades to the Cspk rate was small and did not differ between reward conditions (*Figure 3—figure supplement 2E,F*). We also ruled out the possibility that differences in saccade velocity or direction could explain our results (not shown).

## Complex spike coding of target motion does not depend on reward size

Overall, these results indicate that the Cspk rate differentiates between reward sizes when reward information is first made available, but not during delivery. However, Cspks are also tuned to the direction of target motion (*Kobayashi et al., 1998*; *Stone and Lisberger, 1990*). According to the error signal model, this tuning is a result of image motion on the retina that is caused by the mismatch between target and eye motion (retinal slip). Our sample contained cells that were directionally tuned and not cue responsive (21 cells, example in *Figure 4—figure supplement 1A–C*), cells

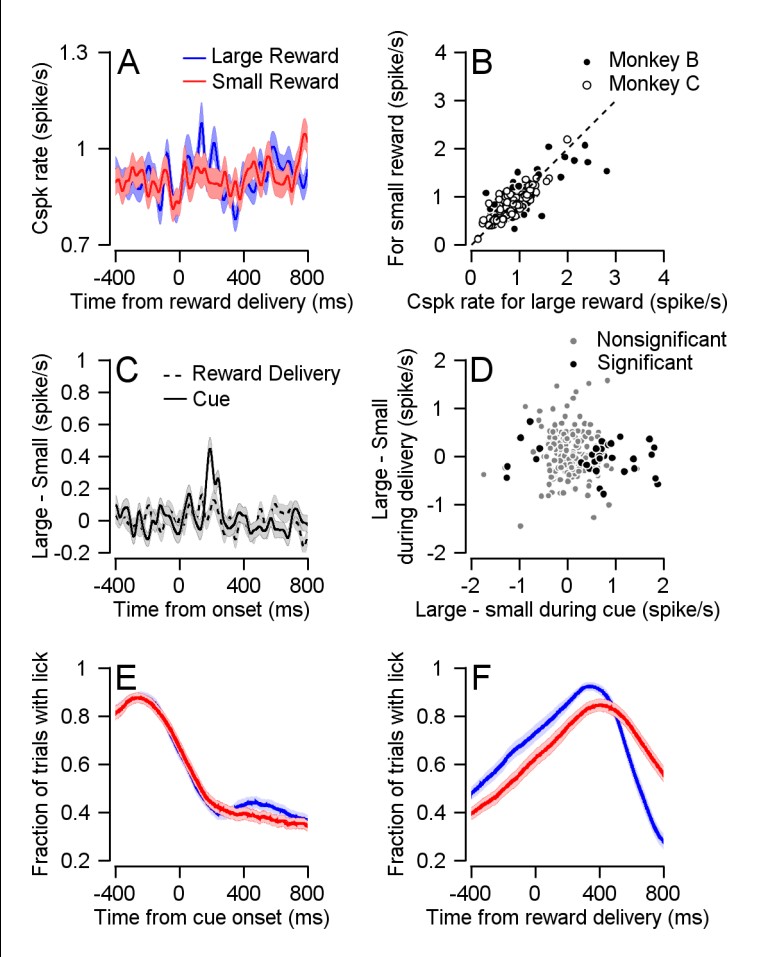

**Figure 3.** Cspk is not modulated by reward size during reward delivery. (**A**) Population PSTHs for different reward conditions aligned to reward delivery. (**B**) Each dot represents the average Cspk rate of an individual cell 100–300 ms large (horizontal) and small (vertical) reward delivery (Signed-rank, Monkey B: p = 0.339, n = 148; Monkey C: p = 0.719, n = 72). (**C**) The differences between the PSTH for large and small rewards aligned to cue or to reward delivery. (**D**) Each dot represents the average Cspk rate of an individual cell 100–300 ms after the cue (horizontal) and reward delivery (vertical; Spearman correlation of all cells: r = −0.069, p = 0.304, n = 220; Spearman correlation of cells that responded to reward size during cue: r = −0.056, p = 0.727, n = 40). (**E**) and (**F**) Fraction of trials with licks, during cue and reward delivery.
DOI: https://doi.org/10.7554/eLife.46870.007
The following figure supplements are available for figure 3:

**Figure supplement 1.** Licking behavior does not underpin the Cspk rate difference during the cue.
DOI: https://doi.org/10.7554/eLife.46870.008
**Figure supplement 2.** Saccades and microsaccades do not underpin the Cspk rate difference during the cue.
DOI: https://doi.org/10.7554/eLife.46870.009

that were cue responsive and not directionally tuned (28 cells, example in *Figure 4—figure supplement 1D–F*) and cells that were both (12 cells, example in *Figure 4—figure supplement 1G–I*).

To determine how Cspk coding of target direction is affected by reward expectation, we focused on directionally tuned cells (33 cells, *Figure 4A,B*). When we examined the Cspk rate in the preferred direction (PD) of the cell and the direction 180° to it (the null direction), we did not find significant differences in the Cspk rate between reward conditions (*Figure 4C*). We aligned the cells to their PD and calculated a population tuning curve for each reward condition. The tuning curves overlapped and were not significantly different (*Figure 4D*).

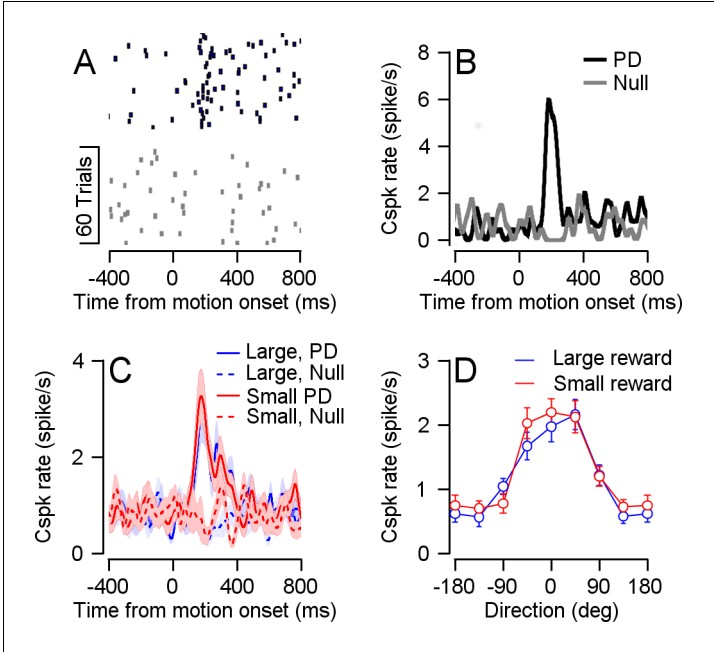

**Figure 4.** Reward did not modulate Cspk direction tuning. (**A**) Raster plot of an example cell in its preferred (black) and null (gray) directions, aligned to target movement onset. (**B**) PSTH of the cell in A. (**C**) Population PSTH for different reward conditions, in the preferred (solid) and null (dashed) directions. (**D**) Population direction tuning curve (Permutation test: p = 0.2156, n = 33).
DOI: https://doi.org/10.7554/eLife.46870.010

The following figure supplements are available for figure 4:

**Figure supplement 1.** Examples of cells Cspk responses to cue and target movement.
DOI: https://doi.org/10.7554/eLife.46870.011

**Figure supplement 2.** Retinal slip due to drift eye movement does not underpin the Cspk rate difference during the cue.
DOI: https://doi.org/10.7554/eLife.46870.012

We also examined the modulation of reward on Cspk rate at different eye velocities. We performed an additional speed task in which we manipulated the target speed (5, 10 or 20 °/s). Eye velocity corresponded to the speed of the target (*Figure 5A*). The effect of expected reward size on eye velocity was evident for all speeds at the average and the single session level (*Figure 5A,B*). Whereas cells responded to the target movement onset (*Figure 5C*), reward expectation did not modulate their response (*Figure 5D*). Together with the directional tuning results, this shows that encoding of reward is limited to the time point at which the reward size is first signaled and not the time when reward drives changes in behavior. Note that the rate of the Cspks did not increase monotonically with target speed (*Figure 5C and D*); we return to this point in the discussion.

Small drift eye movements during fixation can result in retinal slip (*Figure 4—figure supplement 2A,B*). Since Cspks respond to slow visual motion (*Guo et al., 2014*; *Hoffmann and Distler, 1989*), it is possible that the reward size modulation during the cue arose from a retinal slip driven by the appearance of the cue. However, since many of the cells that responded to the cue did not respond during target motion (24 cells), this does not seem to have been the case. Furthermore, we could not find a relationship between drift size or direction and the occurrence of Cspks. Specifically, the drift was similar between trials with and without a Cspk (*Figure 4—figure supplement 2C,D*). Aligning the drift following the cue to the occurrence of a Cspk resulted in a flat line around zero (*Figure 4—figure supplement 2E*), indicating that Cspks were not preceded by increased retinal slip. Finally, when calculating the Cspk responses to the cue separately for trials in which the drift was in the preferred or the null direction of the cell we observed no differences (*Figure 4—figure supplement 2F*).

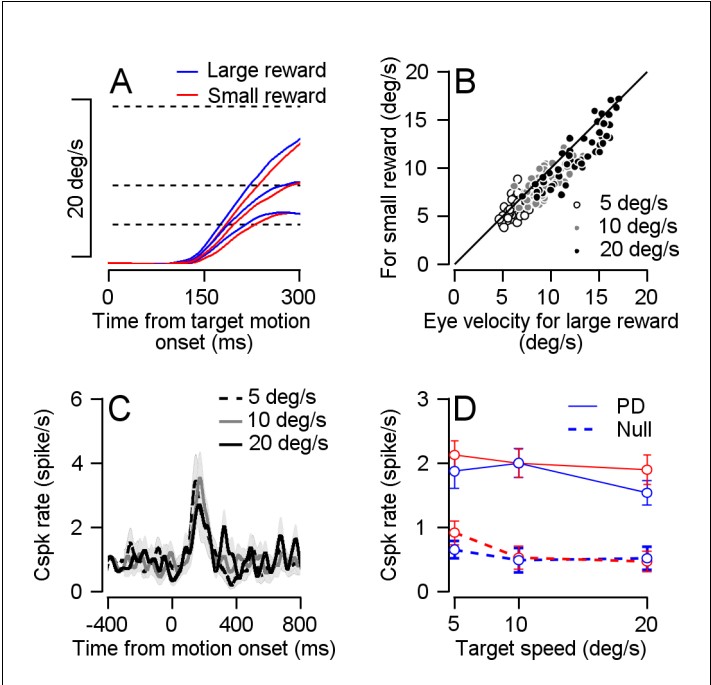

**Figure 5.** Cspk rate was not modulated by reward size at target motion onset in the speed tuning task. (**A**) Average eye velocity traces for experiments in which the color cue signaled a large (blue) or small (red) reward and the target speed was 5 °/s, 10 °/s and 20 °/s. Slower traces correspond to slower target speeds. Dotted lines represent target velocity. (**B**) Individual session average eye velocity 250 ms after target movement onset for large (horizontal) and small (vertical) reward, in the different target velocity conditions (Signed-rank: p = 6*10$^{-16}$, n = 56). (**C**) population PSTHs of cells in their PD for the different speed conditions. (**D**) Population speed tuning curve in the PD (solid) and null (dashed) directions (Permutation test: p = 0.4541, n = 16).
DOI: https://doi.org/10.7554/eLife.46870.013

## The relationship between simple and complex spikes is different for reward and direction tuning

Given that Cspks were modulated by reward size following cue presentation, we went on to examine the Sspk modulations that occur concurrently. Preparatory activity following cues that predict reward or movement had been found in the cerebellum both at the level of the inputs that modulate Sspk rate (*Wagner et al., 2017*) and at the level of their output (*Chabrol et al., 2019*; *Gao et al., 2018*). Recently it was shown that Sspk rate decreases when behavior leads to a reward (*Chabrol et al., 2019*). Within the cells we recorded, Sspk responses to cue presentation were heterogeneous (*Figure 6*, examples in A-C). We found some cells that elevated their Sspk rate in the large versus small reward conditions (*Figure 6A*), others where activity was lower in the large reward condition (*Figure 6B*) and cells in which responses were similar in the large and small reward conditions (*Figure 6C*). Overall, we found more cells in which the Sspk rate was larger for the large reward condition (*Figure 6D*, blue line). However, in a substantial number of cells the Sspk rate was larger for the small reward (*Figure 6D*, red line). As a result of the opposite modulation, at the population level, the difference in Sspk between large and small reward mostly averaged out (*Figure 6E,F*).

The directionally tuned Cspk signal has been linked to the coding of visual errors that instruct motor learning (*Medina and Lisberger, 2008*; *Nguyen-Vu et al., 2013*) by changing the Sspk response to parallel fiber inputs. Cspks generate plasticity in parallel fiber synapses leading to a decrease in the Sspk rate (*Ekerot and Kano, 1985*). This plasticity is thought to underlie the opposite modulations of simple and complex spike rates on different tasks (*Badura et al., 2013*; *Gilbert and Thach, 1977*; *Stone and Lisberger, 1990*). The consistently larger response to the larger reward in the Cspk (*Figure 2*) versus the heterogeneous Sspk response (*Figure 6*), suggests

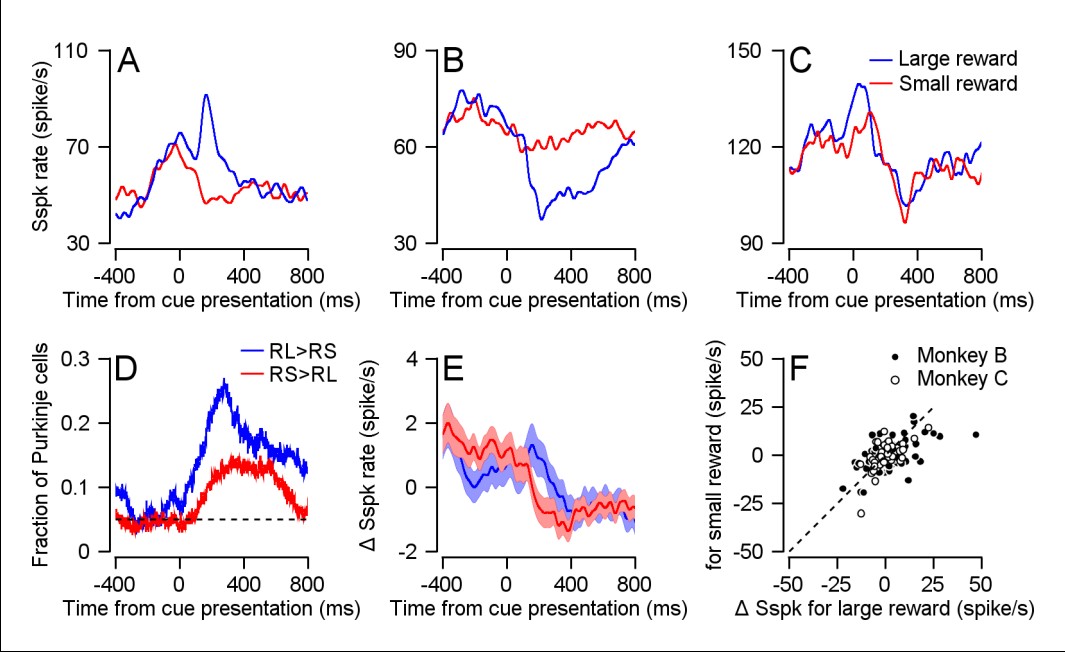

**Figure 6.** Sspk modulations following cue presentation. (**A-C**) Examples of cells' Sspks responses to cue presentation in each reward condition. (**D**) Fraction of cells with a higher Sspk rate in the large reward condition (blue) or small reward condition (red) as a function of time. The dashed line represents the 0.05 false positive chance level. (**E**) Population PSTH, the average Sspk rate of each cell was subtracted. (**F**) Each dot represents the average Sspk rate of an individual cell 100–300 ms following large (horizontal) and small (vertical) reward delivery (Signed-rank, Monkey B: p = 0.142, n = 155; Monkey C: p=0.09, n = 75).
DOI: https://doi.org/10.7554/eLife.46870.014

that the expected opposite modulation between Cspk and Sspk found in relation to movement does not hold for reward related signals.

To test the relationship between Cspks and Sspks directly we compared the rate modulation in the same cell. In our sample of cells, we found the expected opposite modulations during movement. When we aligned the Cspk tuning curve to the preferred direction of the Sspks of the same cell, we found that the Cspk rate decreased in directions for which the Sspk rate increased (*Figure 7A*). To examine whether this effect existed at the single cell level, we calculated the signal correlation for the complex and simple spikes which we defined as the correlation between simple and complex direction tuning curves. We found that most signal correlations were negative; in other words, the Cspks and Sspks were oppositely modulated during movement in most cells (*Figure 7B*). This effect disappeared when we shuffled the phase of the Cspk tuning curve or assigned direction labels randomly (see Materials and methods).

Unlike movement related modulation, the complex and simple spikes were not oppositely modulated following cue presentation (*Figure 7C,D*). If reward-related modulations in Cspks drive Sspk attenuation, we would expect that the higher Cspk rate in the large reward condition would result in a stronger attenuation of Sspks. This would lead to a negative correlation between the complex and simple spike reward modulations during the cue. However, we found that simple and complex spike modulations following cue presentation were uncorrelated (*Figure 7C*). As we observed cells that changed their Sspk rate after the cue without differentiating between reward conditions, we also calculated the correlation between Cspk reward condition modulations and the change in Sspk rate following the cue. In this case as well, we did not find any correlation (*Figure 7D*). Further, the correlations were not significantly different from zero whether we analyzed the full population or only those cells whose Cspks were significantly tuned to reward size during the cue. Thus, the way the difference in Cspk rate during cue affects Sspk encoding and behavior may differ from the one suggested by the error signal model.

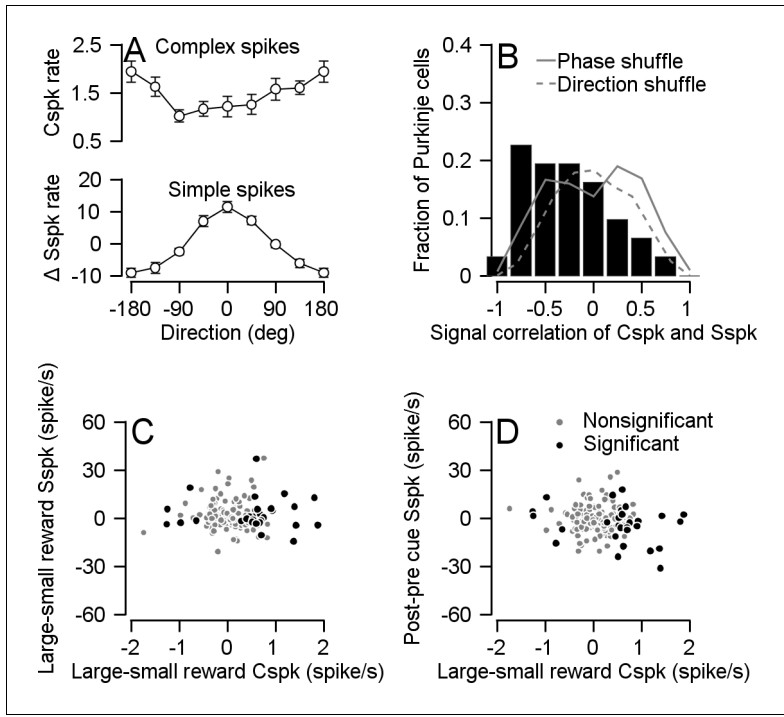

**Figure 7.** Cspk rate negatively correlated with Sspk rate during movement but not during cue presentation. (**A**) Population tuning curve of Cspks (up) and Sspks (bottom), both aligned to the preferred direction of Sspks (Spearman r = −0.3087, p = 7*10$^{-7}$, n = 31). (**B**) Histogram of signal correlations of simple and complex spikes in the population. Solid and dashed lines show the correlations for phased and direction shuffled data (Signed-rank: p=0.002, n = 31). (**C**) Each dot shows individual cell differences in average rate between reward conditions 100–300 ms after cue, in Cspks (horizontal) and Sspks (vertical; Spearman correlation of all cells r = −0.07, p = 0.32, n = 172; Spearman correlation of cells that responded to reward size during cue: r = −0.003, p = 0.98, n = 30) (**D**) Similar to **C** the horizontal position of each dot shows individual cell differences in average Cspk rate between reward conditions 100–300 ms after cue. The vertical axis shows the difference in Sspk firing rate in the time window 100–300 ms after the cue and 100–300 ms before the cue (vertical; Spearman correlation of all cells r = −0.03, p = 0.63, n = 172; Spearman correlation of cells that responded to reward size during cue: r = −0.19, p = 0.31, n = 30).

DOI: https://doi.org/10.7554/eLife.46870.015

## Discussion

The difference in Cspk rate during cue presentation and the lack of difference during reward delivery and pursuit behavior implies that Cspks can act as a reward prediction signal. This finding diverges from the accepted error signal model. The coding of predictive stimuli has been reported in Cspks in the context of error-based learning (*Ohmae and Medina, 2015*). Together with the current results, this suggests a more general role for the cerebellum in associative learning, when learning is both error and reward based (*Heffley and Hull, 2019*; *Kostadinov et al., 2019*; *Thoma et al., 2008*; *Wagner et al., 2017*). The similar Cspk response to the different reward sizes during reward delivery implies that the Cspk coding of reward is not related to reward consumption behavior and does not represent the successful completion of the trial. As we did not observe a reward effect on Cspk rate during pursuit eye movements, when the retinal slip was the largest, our results do not support reward modulation of the Cspk error signal.

### Plasticity and learning from rewards in the cerebellum

Error-based models of the cerebellum link the cerebellar representation of movement, plasticity mechanisms and learning. In this framework, the behavioral command of the cerebellar cortex in response to a stimulus is represented by the Sspk rate of Purkinje cells. Cspks lead to a reduction in the synaptic weight in recently active parallel fibers and thereby change the Sspk rate in response to

similar parallel fiber input (*Ekerot and Kano, 1985*). This change in the Sspk rate is hypothesized to alter the behavioral response to the same stimulus. Thus, when errors occur, the behavior that led to them is eliminated. The same logic cannot apply to learning from rewards since reward strengthens rather than eliminates the behavior that led to the reward (*Thorndike, 1898*).

Consistent with this reasoning, we found that reward-related modulation of Cspks did not exhibit the classical decrease in Sspk activity associated with Cspk activity (*Figure 7*). This result suggests that on our task, other plasticity rules might mask or override the depression. Research on the cerebellum has identified many other sites in which plasticity might drive changes in neuronal activity (*Gao et al., 2012*; *Jörntell and Ekerot, 2002*). Furthermore, the Cspk dependent plasticity in the parallel fibers might also change sign as a result of the network state (*Rowan et al., 2018*). Thus, our results suggest that such mechanisms are engaged when Cspks are modulated by reward.

The Cspk reward signal does not seem to affect cerebellar computation through the same relatively well-understood mechanisms of the Cspk error signal. We also did not find an effect of reward on the Cspk signal during behavior. Thus, the influence of the Cspk reward signal to behavior remains unclear. Moving beyond the level of representation to a mechanistic understanding of the effect of the Cspk reward signal on cerebellar computation and behavior is a crucial next step.

## Relationship to previous studies of the smooth pursuit system

A further demonstration of the existence of independent mechanisms for learning from reward and sensory errors emerges when combining the current results with our recent behavioral study (*Joshua and Lisberger, 2012*). In that study, monkeys learned to predict a change in the direction of target motion by generating predictive pursuit movements. The size of the reward did not modulate the learning process itself but only the execution of the movement (*Joshua and Lisberger, 2012*). The critical signal for direction change learning has been shown to be the directionally tuned Cspk signal (*Medina and Lisberger, 2008*). Our findings that the target direction signal is not modulated by reward provides a plausible explanation at the implementation level for this behavioral finding. The directionally tuned Cspks that drive learning are not modulated by reward; therefore, learning itself is reward independent.

In the current study, the Cspk rate did not increase with target speed (*Figure 5*). At least one study has reported a monotonic increase between Cspk rate and motion speed (*Kobayashi et al., 1998*). The specific experimental protocol we used might have led to the lack of speed coding. The vast majority of trials in which the monkeys were engaged were at 20 °/s, and we only measured responses at different speeds in a minority of the sessions (see Materials and methods). Therefore, it is possible that the monkey developed a speed prior (*Darlington et al., 2018*) and hence was expecting the target to move at 20 °/s. Violation of this prior in the slower motion trials might have potentiated the response and masked the speed tuning. Behavioral support for such a prior comes from the eye speed response to low speed targets (5 °/s) in which the eye speed overshot the target speed (*Figure 5A,B*). Other possibilities such as the recorded population or the properties of the visual stimuli might also have contributed to the lack of speed tuning.

## Future directions

The reward signal we found is similar to reward expectation signals in dopaminergic neurons of the ventral tegmental area (VTA) and substantia nigra pars compacta (*Schultz et al., 1997*). The VTA projects to the inferior olive (*Fallon et al., 1984*) and recently, direct projections from the cerebellum to dopaminergic neurons in the VTA have been found (*Carta et al., 2019*). Reward signals have also been found in cerebellar granular cells that modulate the Sspk rate in Purkinje cells (*Wagner et al., 2017*) and in the deep cerebellar nuclei (*Chabrol et al., 2019*). Researching the differences and interactions of reward signals is an important next step in understanding how reward is processed. In particular, future research will need to investigate the source of the reward information in the inferior olive.

Another interesting question is whether the Cspk representation of reward depends on the range of possible rewards. Our results demonstrate that the Cspk rate is informative of future reward size. Expected reward size might be represented in the cerebellum in an absolute manner, based on its physical size, or in relative order, based on its motivational value in comparison to other available rewards (*Cromwell et al., 2005*; *Tremblay and Schultz, 1999*). Our results show that when a small

reward cue is presented, there is no increase in the Cspk rate (*Figure 2D*). Although this cue predicts a future reward, it does not elicit a Cspk response. This hints that the Cspk representation may be relative and not absolute. To further verify this, we need to construct a task in which we examine the same reward size in different contexts.

## Conclusion

To sum up, the current study demonstrates that a population of Purkinje cells receive a reward predictive signal from the climbing fibers. Our results show that the reward signal is not limited to the direct rewarding consequences of the behavior. These results thus suggest that the cerebellum receives information about future reward size. Our results go beyond previous findings of cerebellar involvement in the elimination of undesired behavior, to suggest that the cerebellum receives the relevant information that could allow it to adjust behavior to maximize reward.

## Materials and methods

We collected neural and behavioral data from two male Macaca Fascicularis monkeys (4–5 kg). All procedures were approved in advance by the Institutional Animal Care and Use Committees of the Hebrew University of Jerusalem and were in strict compliance with the National Institutes of Health Guide for the Care and Use of Laboratory Animals. We first implanted head holders to restrain the monkeys' heads in the experiments. After the monkeys had recovered from surgery, they were trained to sit calmly in a primate chair (Crist Instruments) and consume liquid food rewards (baby food mixed with water and infant formula) from a tube set in front of them. We trained the monkeys to track spots of light that moved across a video monitor placed in front of them.

Visual stimuli were displayed on a monitor 45 cm from the monkeys' eyes. The stimuli appeared on dark background in a dimly lit room. A computer performed all real-time operations and controlled the sequences of target motions. The position of the eye was measured with a high temporal resolution camera (1 kHz, Eye link - SR research) and collected for further analysis. Monkeys received a reward when tracking the target successfully.

In subsequent surgery, we placed a recording cylinder stereotaxically over the floccular complex. The center of the cylinder was placed above the skull targeted at 0 mm anterior and 11 mm lateral to the stereotaxic zero. We placed the cylinder with a backward angle of 20° and 26° for monkey B and C respectively. Quartz-insulated tungsten electrodes (impedance of 1–2 Mohm) were lowered into the floccular complex and neighboring areas to record simple and complex spikes using a Mini-Matrix System (Thomas Recording GmbH). When lowering the electrodes, we searched for neurons that responded during pursuit eye movements (see direction task) but often collected data from neurons that did not respond to eye movements. Overall, we recorded complex spikes from 148 and 72 neurons from monkeys B and C respectively. Of these, the Sspks of 28 and 19 neurons from monkeys B and C were directionally tuned during the direction task (Kruskal-Wallis test, $\alpha = 0.05$).

Signals were digitized at a sampling rate of 40 kHz (OmniPlex, Plexon). For the detailed data analysis, we sorted spikes offline (Plexon). For sorting, we used principal component analysis and corrected manually for errors. In some of the cells the Cspks had distinct low frequency components (*Warnaar et al., 2015*; *Zur and Joshua, 2019*; for example *Figure 1—figure supplement 2B*, left column and *Figure 2—figure supplement 1*). In these cells, we used low frequency features to identify and sort the complex spikes. We paid special attention to the isolation of spikes from single neurons. We visually inspected the waveforms in the principal component space and only included neurons for further analysis when they formed distinct clusters. Sorted spikes were converted into timestamps with a time resolution of 1 ms and were inspected again visually to check for instability and obvious sorting errors.

We used eye velocity and acceleration thresholds to detect saccades automatically and then verified the automatic detection by visual inspection of the traces. The velocity and acceleration signals were obtained by digitally differentiating the position signal after we smoothed it with a Gaussian filter with a standard deviation of 5 ms. Saccades were defined as an eye acceleration exceeding 1000 °/s$^2$, an eye velocity crossing 15 °/s during fixation or eye velocity crossing 50 °/s while the target moved. To calculate the average of the smooth pursuit initiation we first removed the saccades and treated them as missing data. We then averaged the traces with respect to the target movement direction. Finally, we smoothed the traces using a Gaussian filter with a standard deviation of 5 ms.

We also recorded licking behavior to control for behavioral differences between reward conditions that might confound our results. Licks were recorded using an infra-red beam. Monkey B tended not to extend its tongue, therefore we recorded lip movements.

## Experimental design

### Direction task

Each trial started with a bright white target that appeared in the center of the screen (*Figure 1A*). After 500 ms of presentation, in which the monkey was required to acquire fixation, a colored target replaced the fixation target. The color of the target signaled the size of the reward the monkey would receive if it tracked the target. For monkey B we used blue to signal a large reward (~0.2 ml) and red to signal a small reward (~0.05 ml); for monkey C we used yellow to signal a large reward and green to signal a small reward. After a variable delay of 800–1200 ms, the targets stepped in one of eight directions (0°, 45°, 90°, 135°, 180°, 225°, 270°, 315°) and then moved in the direction 180° from it (step-ramp, *Rashbass and Westheimer, 1961*). For both monkeys, we used a target motion of 20 °/s and a step to a position 4° from the center of the screen. The target moved for 750 ms and then stopped and stayed still for an additional 500–700 ms. When the eye was within a $3 \times 3$ degree window around the target the monkey received a juice reward.

### Speed task

During the direction task we online fitted a Sspk tuning curve for each cell and approximated the cell's PD. If a cell seemed directionally tuned, we ran an additional speed task. The temporal structure of the speed task was the same as the direction task. The step size was set to minimize saccades and was 1°, 2° and 4° for a target speed of 5, 10 or 20 °/s. The targets could move either in the approximate PD of the cell or the direction 180° from it, which we termed the null direction. The targets moved at 5, 10 or 20 °/s.

### Choice task

Monkeys were required to choose one of two targets (large or small reward) presented on the screen (*Figure 1—figure supplement 1A*). We used this task to determine whether the monkeys correctly associated the color of the target and the reward size (*Figure 1—figure supplement 1B*). Their choice determined the amount of reward they received. Each trial began with a 500 ms fixation period, similar to the tasks described previously. Then two additional colored spots appeared at a location eccentric to the fixation target. One of the colored targets appeared 4° below or above the fixation target (vertical axis) and the other appeared 4° to the right or left of the fixation target (horizontal axis). The monkey was required to continue fixating on the fixation target in the middle of the screen. After a variable delay of 800–1200 ms, the white target disappeared, and the colored targets started to move towards the center of the screen (vertically or horizontally) at a constant velocity of 20 °/s. The monkey typically initiated pursuit eye movement that was often biased towards one of the targets (*Figure 1—figure supplement 1C*). After a variable delay, the monkeys typically made saccades towards one of the targets. We defined these saccades as an eye velocity that exceeded 80 °/s. The target that was closer to the endpoint of the saccade remained in motion for up to 750 ms and the more distant target disappeared. The monkey was required to track the target until the end of the trial and then received a liquid food reward as a function of the color of the target.

## Data analysis

All analyses were performed using Matlab (Mathworks). When comparing reward conditions, we only included cells that were recorded for a minimum of 20 trials (approximately 10 for each condition). When performing analyses that included additional variables such as target direction or velocity, we set a minimum of 50 trials (approximately 3–4 for each condition).

To study the time varying properties of the response, we calculated the PSTH at a 1 ms resolution. We then smoothed the PSTH with a 10 ms standard deviation Gaussian window, removing at least 100 ms before and after the displayed time interval to avoid edge effects. Note that this procedure is practically the same as measuring the spike count per trial in larger time bins. We defined cells that responded significantly differently to reward conditions during the cue using the rank-sum test on the mean number of spikes 100–300 ms after cue onset.

To calculate the tuning curves, we averaged the responses in the first 100–300 ms of the movement. We calculated the preferred direction of the neuron as the direction that was closest to the vector average of the responses across directions (direction of the center of mass). We used the preferred direction to calculate the population tuning curve by aligning all the responses to the preferred direction. We defined a cell as directionally tuned if a one-way Kruskal-Wallis test (the case of 8 directions, directions task), or a rank-sum test (the case of two directions, speed task), revealed a significant effect for direction. We present reward modulation on movement parameters only for directionally tuned cells and also confirmed that if we took the full population there was no reward modulation at motion onset (Signed-rank: Monkey B, $p = 0.8904$, $n = 148$; Monkey C, $p = 0.4487$, $n = 72$).

To statistically test the significance of the effect of reward direction tuning we used a permutation test. We first calculated separate tuning curves for each cell in the two reward conditions. We then chose a random subset of combinations of cells and directions and reversed the small and large reward labels of this subset. We then calculated the population PSTHs for the shuffled 'small' and 'large' reward conditions. Our statistic was the mean square distance of the two tuning curves. We used the percentile of the statistic of the unshuffled data to calculate the p-value. We used a similar test for the speed task in which the subset we chose was a random combination of cell, direction and speed.

We calculated the fraction of cells whose Sspk rate was different between reward conditions as a function of time (*Figure 6D*) by using left and right-tailed rank-sum tests on a moving time window. For each cell, we looked for time points in which there were significantly more Sspks in the large reward trials in comparison to the small (RL > RS) and time points in which there were significantly more Sspks in the small reward trials in comparison to the large (RS > RL). We tested each time point by calculating the number of Sspks in each trial in time bins of 200 ms surrounding it. We then tested if the number of Sspks in large reward trials was significantly different using both left and right signed-rank test. We classified that time point as RL > RS, RS > RL or neither according to the result of the tests. We then calculated the fraction of cells in each category for every time point.

We calculated the signal correlation of each cell's Cspks and Sspks by calculating a tuning curve of each spike type and computing the Pearson correlation of the tuning curves (*Figure 7B*). As a control, we performed the same analysis on shuffled data. In the phase shuffled control, we shuffled the Cspk tuning curves by different phases while preserving their relative order. For example, shuffling by a phase of 45° meant moving the response at 0° to 45°, 45° to 90°, 315° to 0° and so on. In the direction shuffle, we assigned random direction labels to the Cspk responses.

We calculated the cross-correlation of complex and simple spikes (*Figure 1—figure supplement 2D*) by calculating the PSTH of Sspks aligned to a Cspk event. We removed Cspks that occurred less than 100 ms after the trial began or less than 100 ms before a trial ended since we did not have sufficient information to calculate the PSTH. We manually removed spikes that were detected 1 ms before a Cspk or 2 ms after, because occasionally they could not be distinguished from Cspk spikelets.

To control for the direct responses to licking we approximated the contribution of the Cspk response to licking (*Figure 3—figure supplement 1D*) to the Cspk response to cue. We first calculated the peri-event time histogram (PETH) of each cell aligned to lick onset without separating the reward conditions (*Figure 3—figure supplement 1C*). Then, for every trial, we created synthetic data in which the firing rate around each lick onset was set to the average lick triggered PETH. Firing rates during times that were outside the range of the PETH (300 ms) were treated as missing data. We then averaged these single trial estimations of the firing rate to calculate the predicted PSTH for each reward condition, aligned to cue presentation. We performed a similar analysis for lick offset (*Figure 3—figure supplement 1D*, dashed line) and saccades (*Figure 3—figure supplement 2F*).

To control for the response to retinal slip following cue presentation, we calculated the vertical and horizontal drift. Drift velocity during fixation is small and thus eye tracker measurements of drift movements are prone to measurement noise. We noted that a large fraction of the variance in drift position is explained by changes in pupil size ($R^2$ vertical median = 0.94, $R^2$ horizontal median = 0.17, $n = 208$; *Kimmel et al., 2012*). To examine differences in the drift between reward conditions independently of pupil size, we fitted a linear model between pupil size and eye position for the averages of each recording session. We subtracted the position predicted by pupil size from the measured position for each trial and performed further analyses on these corrected traces.

We did not correct for multiple comparisons in our analysis. We either used a small number of tests over the entire population or a large number of tests on individual cells that were only used as a criterion (for example, whether a cell differentiated between reward conditions during the cue). When using a test as a criterion we did not infer the existence of responsive cells but rather used it to classify cells into subpopulations.

## Acknowledgements

We thank Y Botschko for technical assistance. This study was supported by a HFSP career development award, the Israel Science Foundation and the European Research Council.

## Additional information

### Funding

| Funder | Grant reference number | Author |
| --- | --- | --- |
| H2020 European Research Council | imove 755745 | Mati Joshua |
| Human Frontier Science Program | CDA 00056 | Mati Joshua |
| Israel Science Foundation | 38017 | Mati Joshua |

The funders had no role in study design, data collection and interpretation, or the decision to submit the work for publication.

### Author contributions

Noga Larry, Conceptualization, Formal analysis, Investigation, Visualization, Methodology, Writing—original draft, Writing—review and editing; Merav Yarkoni, Conceptualization, Data curation, Investigation, Methodology, data acquisition; Adi Lixenberg, Data curation, data acquisition; Mati Joshua, Conceptualization, Resources, Data curation, Formal analysis, Supervision, Funding acquisition, Validation, Investigation, Visualization, Methodology, Writing—review and editing

### Author ORCIDs

Noga Larry ⓘD https://orcid.org/0000-0001-8750-2182
Mati Joshua ⓘD https://orcid.org/0000-0003-2602-3334

### Ethics

Animal experimentation: All the procedures described in this paper were approved in advance by the Institutional Animal Care and Use Committees of the Hebrew University of Jerusalem (ethics approval number MD15145854) and were in strict compliance with the National Institutes of Health Guide for the Care and Use of Laboratory Animals.

### Decision letter and Author response

Decision letter https://doi.org/10.7554/eLife.46870.020
Author response https://doi.org/10.7554/eLife.46870.021

## Additional files

### Supplementary files

• Transparent reporting form DOI: https://doi.org/10.7554/eLife.46870.016

### Data availability

The data used in this paper is available in: https://github.com/MatiJlab.

The following dataset was generated:

| Author(s) | Year | Dataset title | Dataset URL | Database and Identifier |
|---|---|---|---|---|
| Larry N, Yarkoni M, Lixenberg A, Joshua M | 2019 | MatiJlab | https://github.com/Ma-tiJlab | GitHub, MatiJlab |

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
