## [Decision Letter]

Thank you for submitting your article "Cerebellar climbing fibers encode expected reward size" for consideration by *eLife*. Your article has been reviewed by Ronald Calabrese as the Senior Editor, a Reviewing Editor, and three reviewers. The following individuals involved in review of your submission have agreed to reveal their identity: Steve A Edgley (Reviewer #2).

The reviewers have discussed the reviews with one another and the Reviewing Editor has drafted this decision to help you prepare a revised submission.

Summary:

This paper presents the dramatic finding that cerebellar Purkinje cell complex spikes (CS), which are generally thought to carry motor error signals, also show activity related to a cue indicating the reward value of successful performance of a future motor task. The consensus view of CS function suggests that they represent precisely timed instructive signals which act to modify cerebellar cortical circuits in order to optimise cerebellar outputs and thus motor performance. In this paper, complex spikes were recorded from Purkinje cells of monkeys performing a smooth-pursuit task. Before the pursuit target is presented, a visual cue predicts the value of a reward contingent upon good motor performance following a "go" signal after a variable delay. The key finding is that CSs are powerfully modulated with precise timing shortly after the cue signalling the size of the future reward. This modulation is dramatically different for large reward predicting cues than for small reward predicting cues. The results are unexpected and the authors show convincingly that the CS are related to the reward predicting cue rather than to the subsequent reward. High reward predicting cues are followed by faster movements, so the cue related CS signal correlates with that. Similar reward predicting cue-related modulation is seen in both cells with target/eye-movement tuned CSs and non-target/eye movement related CSs. A further remarkable property is that responses to large reward predicting cues increase complex spike frequency whereas small reward predicting cues decrease it, suggestive of ideas from temporal difference (TD) learning.

This is a big finding for the field and would fit nicely with other recent work on reward signals in the cerebellum and connections between the cerebellum and dopamine system. Similar cue -related activity has been described previously for limb motor tasks (prefrontal cortex (Tsujimoto et al., 2012), anterior striatum/SMA (Romoand Schultz, 1992) and for saccade-related frontal lobe areas (Roesch and Olson, 2003), as well as dopaminergic substantia nigra/VTA neurons, but not previously in CSs. There was considerable enthusiasm about this work, and also some issues that should be addressed.

Essential revisions:

1) Additional effort should be made to address the possibility of "lurking" variables, such as movements, that could explain the reward-related CS activity rather than reward per se. It is likely that there are many motor responses related to orienting that may be correlated with "reward prediction. The analyses presented in Figure 3E and F or Supplementary figure 3 are not sufficient to rule out these potential confounds. While it would be unreasonable to expect the authors to control for all of them, additional exposition of some of the most likely should be possible through additional analyses of data already on hand and additional discussion of this issue. This is especially relevant since the authors did not do the experiments typically required to show a TD-like signal (e.g. reward omission) as has been reported in dopamine neurons (Schultz) and aversive CS signals (Ohmae and Medina, 2015). In particular, the following would improve the claim that the CS response can be attributed to differences in predicted reward size.

a) More extensive analysis of licking would be helpful.

Figure 3E,F plot the fraction of trials with lick? What about the number or timing of licks? Also, while it is true, as the authors state, that the monkey's licking rates are very high before the cue and decrease around the time of the cue, there does appear to be a substantial difference in licking behavior for large and small reward trials following the cue--around the time of the "reward prediction" CS. As IO neurons tend to depress during prolonged activation, it's perhaps not surprising that they would not be strongly driven during the pre-cue licking bouts, especially if the licks are not time locked to a particular event (which would cause jitter that could "blur" out consistent spiking). Further, the authors never show what the CSs are doing more than 200 ms preceding the cue (when licking is maximal).

b) The analysis of saccades was not especially compelling. An advantage of doing this study in the floccular complex is that we know that complex spikes encode retinal slip, so for the neurons with directionally tuned CSs there is one key potential lurking variable, which is readily addressed, and that is eye movements that could produce retinal slip in the preferred direction of the CSs-i.e., eye movements in the opposite direction. More extensive analysis of the direction and velocity of eye movements during the cue period is needed.

c) A key question is whether there is any cue-related SS modulation – might throw light on the function of the Purkinje cells that were not tuned for visual target direction and on events around the time of the cue related to the complex spikes.

2) Fuller presentation of the results. In general, the presentation of the results was clear, but to fully understand the details, readers must flit between the Results section, figures and Materials and methods section to try to follow some of the numbers and the specifics of the task.

a) Shortly after the cue presentation the data from Figure 2A suggests that the instantaneous discharge rate of complex spikes reaches around 4 Hz. Does this include multiple complex spikes with short Interspike intervals on a single trial? I strongly feel that an illustration of raw data would be very helpful here to allow complex spike and simple spike waveforms to be compared, and that this should have a broader timescale than in Figure 2. For example, this would make a good supplementary data figure.

b) The description of the task in subsection “Complex spikes encode the size of the expected reward” does not fully describe what happens: the coloured target steps in one direction before moving linearly in the direction orthogonal to that direction. This is buried at the end of the paper in the Materials and methods section, and I suspect a very small proportion of the readers will look at that. It really needs to be before or at the time that the task is described.

c) An issue is that the recordings were sampled from a heterogeneous population of Purkinje cells from the flocculus. The Materials and methods section says there were 149 cells, Figure 2 148.

d) Scatter plots in Figure 2D shows some complex spikes with very low rates in the 100-300 ms interval for both small and large rewards: this could arise if only very few trials were averaged. I cannot see data on minimum numbers of trials in the paper – it should be made available.

e) Differentiation is made in Figure 3D between individual Purkinje cells with significant complex spike modulation to the cue and those in which there was not a significant difference. These numbers need to be available in the manuscript – ideally in subsection “Complex spikes encode the size of the expected reward”. At present, it is buried in the Materials and methods section at the end.

f) Following from this, an example cell is shown in Figure 4 which had both cue-related reward modulation and directional modulation. Again, some numbers on these would be valuable; how many of the sample were significantly modulated by both?

g) Figure 6 compares the complex and simple spike modulations, but does not say anything about simple spike modulation related to the reward predicting the cue. At least a brief mention is needed – were any significantly modulated? Is so, were they different for large and small reward predicting cues?

h) The final paragraph of subsection “Representation of reward and target motion in the population” states that the motion tuned cells add an early response which was not selective to reward, which is clear in Figure 7B. Is this a qualitative description or were these significant differences? Since these are population curves, does this reflect heterogeneity in the response pattern, or is it a genuine representation of the behaviour of individual cells?

i) How were the cross correlations between CS and SS calculated? (subsection “Data analysis”). One might expect that the authors would calculate the correlations of the two types of spikes triggered on the onset of the cue or pursuit or some other trigger related to the task. It seems that instead they have calculated it based on CS triggered SS PSTHs. This is strange since this is typically the way that people verify that the CS is coming from the same Purkinje cell as the SS. Could it be that in CS with low negative correlations with SS the CS and SS are not actually coming from the same Purkinje cell?

j) The MRI images in Figure 1—figure supplement 2 don't do much to help convince readers that the authors are recording in or near the VPFL. At the very least, some labels would help. Even better would be if the imaging can be done with an electrode or similar placed in a recording track (which would hopefully reside in the VPFL).

k) In the Conclusion, the statement that the paper demonstrates "how climbing fibres encode predicting the reward size" is not really appropriate – it shows how a subpopulation of CSs innervating Purkinje cells in the flocculus have activity that encodes predicted reward. I also disagree with the statement that the authors have "found signals that can be used by the cerebellum to drive behaviour that maximises upcoming reward" – that is entirely possible, but is not shown in this paper – may be used would be more appropriate.

3) Additional discussion of the potential function of the CS activity related the reward cue.

It is not clear what causal role the authors believe the "reward prediction" signal is playing in the task the monkeys are performing. The cue-related CSs come at a time when the Purkinje cells are not engaged in movement and relate differently to ongoing SS compared to CS occurring during movement – in this case what is their function? On the one hand, the authors show that the pursuit velocity is higher on the large reward trials; but, on the other hand, they show that the Purkinje cells with the highest "reward prediction" signals are the least tuned for the direction of pursuit. How do the authors think these untuned Purkinje cells fit within the pursuit control system? Do they even have access to the motor neurons controlling the eye? If not, why does the cerebellum need such a "reward prediction" signal during this task? Could the increased velocities on the large reward trials not just as easily be explained by increased attention/motivation that is not necessarily dependent on the cerebellum?

This is especially relevant since the majority of the neurons recorded by the authors did not appear to be within the VPFL, at least as it has been defined by Lisberger and others. That is, the percentage of Purkinje cells with directionally tuned simple spikes is very low compared to what is typically reported for VPFL Purkinje cells and few of the CSs seem to encode retinal slip signals. If the Purkinje cells are not in the VPFL how do the authors know they are even causally involved in the task? More information should be provided about the relative locations of the recorded cells that are tuned and untuned with reference to what is known in the literature about non-visual CS receptive fields in the areas around the flocculus.

A weakness of the task design is that it is impossible to distinguish between quantitative versus relative reward size encoding. Although the authors do not claim to study this directly, it is implicit in the learning models cited that there be quantitative reward size encoding (see point #2).

[Editors' note: further revisions were requested prior to acceptance, as described below.]

Thank you for submitting your article "Cerebellar climbing fibers encode expected reward size" for consideration by *eLife*. Your article has been reviewed by Jennifer Raymond as the Reviewing Editor, and Ronald Calabrese as the Senior Editor.

The Reviewing Editor has drafted this decision to help you prepare a revised submission.

Summary:

The authors have addressed many of the points raised in the last review. However, this central issue still has not been convincingly addressed, namely, whether the complex spikes are responding to the cue for reward size per se, rather than something else correlated with the cue, such as movements the animals may well make when anticipating reward and the associated sensory feedback.

Essential revisions:

To be fair, this issue is also a potential confound for many or all of the reports that have been coming out about reward coding in the cerebellum. The very exciting thing about the present experimental design is that it has the potential to nail this issue in a way that other studies have not been able to. In contrast to much of the work on reward coding in the cerebellum, which has been done in regions of the cerebellum where very little is known about coding, in this study, the recordings were made from the floccular complex of the cerebellum, where the signals carried by the simple spikes and complex spikes are probably better understood than anywhere in the cerebellum. And what is known about coding by the complex spikes is that they encode image motion on the retina, or retinal slip, in a particular, preferred direction (almost always contraversive or up). Therefore, the most important potential confound for the claim that the complex spikes encode upcoming reward size, is the possibility that the expectation of a large reward causes the animal to make smooth eye movements (slow drift) that results in retinal slip in the preferred direction of the climbing fiber. The authors analyzed licking and saccades, but the most important potential 'lurking variable" to worry about is the known responsiveness of complex spikes in this region of cerebellum to retinal slip at speed of a few degress/s, and that was not convincingly analyzed. The good news is that is quite do-able, for example, by doing a complex spike-triggered average of retinal slip, and comparing the results for the period after cue presentation with those during target motion in the "on" direction for the complex spikes. Unfortunately, the number of cells that were both responsive to the cue and directionally tuned (12 cells) may not be sufficient, hence more cells may need to be recorded.

[Editors' note: further revisions were requested prior to acceptance, as described below.]

Thank you for submitting your article "Cerebellar climbing fibers encode expected reward size" for consideration by *eLife*. Your article has been reviewed by a Reviewing Editor and Ronald Calabrese as the Senior Editor.

The Reviewing Editor has drafted this decision to help you prepare a revised submission.

The authors have responded in good faith to the concern about the possibility that retinal slip, which is known to drive climbing fibers in the region of the cerebellum recorded, might contribute to the reward-correlated climbing fiber responses. I appreciate that the measurement noise for the camera-based eye tracking limits what can be done to some extent, and the authors have gone to considerable effort to deal the methodological limits.

The analysis provided in Figure 4—figure suppplement 2B is helpful.

Figure 4—figure supplement 2A,C would be more helpful if the variance was provided as well as the mean. If the mean was zero and the variance was 10deg/s, there would be a lot more image motion in the preferred direction than if the mean was zero and variance was zero. This matters since there is an asymmetry in the ability of preferred ad anti-preferred direction image motion to increase and suppress the climbing fiber firing rate because of the low basal firing rate and floor at zero.

The analysis provided in Figure 4—figure supplement 2D is based on an assumption that is not supported by the literature--that the climbing fiber response to retinal slip increases linearly with the velocity of image motion. On the contrary, in rabbit, cat, and rat, climbing fibers in the flocculus are tuned for low retinal slip speeds, and their responses fall off with speeds >1-2{degree sign}/s (Simpson and Alley, 1974; Blanks and Precht, 1983; Kusunoki et al., 1990; Fushiki et al., 1994). In monkeys, some climbing fibers respond to higher speeds, yet the assumption of a linear increase with image velocity is not supported (Noda et al., 1987; Hoffmann and Distler, 1989; Guo et al., 2014).

It is still not clear to me why the authors would not do the obvious thing of a complex spike triggered average of eye velocity in the preceding ~200 ms (for complex spikes in the period associated with the reward cue). After going through the trouble of correcting for pupil size artifacts in the camera data, why not give the climbing fiber spike-triggered eye velocity average a try? Those measurements might be dominated by noise, yet it would still be nice to know that nothing could be detected. Because if something is detected despite the measurement limitations, that would require a reinterpretation of the main result

Essential revisions:

Remove Figure 4—figure supplement 2, panel D, unless the authors can provide some justification for the assumption of linear increase in climbing fiber response with image velocity.

---

## [Author Response]

Summary:This paper presents the dramatic finding that cerebellar Purkinje cell complex spikes (CS), which are generally thought to carry motor error signals, also show activity related to a cue indicating the reward value of successful performance of a future motor task. The consensus view of CS function suggests that they represent precisely timed instructive signals which act to modify cerebellar cortical circuits in order to optimise cerebellar outputs and thus motor performance. In this paper, complex spikes were recorded from Purkinje cells of monkeys performing a smooth-pursuit task. Before the pursuit target is presented, a visual cue predicts the value of a reward contingent upon good motor performance following a "go" signal after a variable delay. The key finding is that CSs are powerfully modulated with precise timing shortly after the cue signaling the size of the future reward. This modulation is dramatically different for large reward predicting cues than for small reward predicting cues. The results are unexpected and the authors show convincingly that the CS are related to the reward predicting cue rather than to the subsequent reward. High reward predicting cues are followed by faster movements, so the cue related CS signal correlates with that. Similar reward predicting cue-related modulation is seen in both cells with target/eye-movement tuned CSs and non-target/eye movement related CSs. A further remarkable property is that responses to large reward predicting cues increase complex spike frequency whereas small reward predicting cues decrease it, suggestive of ideas from temporal difference (TD) learning.This is a big finding for the field and would fit nicely with other recent work on reward signals in the cerebellum and connections between the cerebellum and dopamine system. Similar cue -related activity has been described previously for limb motor tasks (prefrontal cortex (Tsujimoto et al., 2012), anterior striatum/SMA (Romoand Schultz, 1992) and for saccade-related frontal lobe areas (Roesch and Olson, 2003), as well as dopaminergic substantia nigra/VTA neurons, but not previously in CSs. There was considerable enthusiasm about this work, and also some issues that should be addressed.Essential revisions:1) Additional effort should be made to address the possibility of "lurking" variables, such as movements, that could explain the reward-related CS activity rather than reward per se. It is likely that there are many motor responses related to orienting that may be correlated with "reward prediction. The analyses presented in Figure 3E and F or Supplementary figure 3 are not sufficient to rule out these potential confounds. While it would be unreasonable to expect the authors to control for all of them, additional exposition of some of the most likely should be possible through additional analyses of data already on hand and additional discussion of this issue. This is especially relevant since the authors did not do the experiments typically required to show a TD-like signal (e.g. reward omission) as has been reported in dopamine neurons (Schultz) and aversive CS signals (Ohmae and Medina, 2015). In particular, the following would improve the claim that the CS response can be attributed to differences in predicted reward size.

In the revised manuscript we thoroughly controlled for the licking and eye movements. We added Figure 3—figure supplement 1 and Figure 3—figure supplement 2 that show that licking and eye movements cannot explain the reward modulation at cue presentation

a) More extensive analysis of licking would be helpful.Figure 3E,F plot the fraction of trials with lick? What about the number or timing of licks? Also, while it is true, as the authors state, that the monkey's licking rates are very high before the cue and decrease around the time of the cue, there does appear to be a substantial difference in licking behavior for large and small reward trials following the cue--around the time of the "reward prediction" CS. As IO neurons tend to depress during prolonged activation, it's perhaps not surprising that they would not be strongly driven during the pre-cue licking bouts, especially if the licks are not time locked to a particular event (which would cause jitter that could "blur" out consistent spiking). Further, the authors never show what the CSs are doing more than 200 ms preceding the cue (when licking is maximal).

Licking cannot explain the reward modulation at the cue, for the following reasons. First, we separately calculated the PSTH in each reward condition in trials with and without the initiation of a lick (Figure 3—figure supplement 1A,B). Even when the monkey did not initiate a lick, the difference between reward conditions was significant. Second, we calculated the PETEH of the cells that responded to the cue aligned to the onset and offset of licks (Figure 3—figure supplement 1C). We found that the lick response was small. Third, we used this small response together with the minor differences in the behavior between large and small rewards to show that the contribution of licking to the Cspk response was negligible (Figure 3—figure supplement 1D). Finally, to control for the temporal aspects of the licking we performed the analysis on both onset and offset of the licking.

As requested, we also display more time before cue presentation in Figure 2A to show that the Cspk rate did not increase when licking was maximal before the cue.

b) The analysis of saccades was not especially compelling. An advantage of doing this study in the floccular complex is that we know that complex spikes encode retinal slip, so for the neurons with directionally tuned CSs there is one key potential lurking variable, which is readily addressed, and that is eye movements that could produce retinal slip in the preferred direction of the CSs-i.e., eye movements in the opposite direction. More extensive analysis of the direction and velocity of eye movements during the cue period is needed.

We now further address this point by analyzing saccades during the cue period. We calculated the fraction of trials with saccades at different time points along the trial (Figure 3—figure supplement 2A,B). Only a minority of trials had saccades during the cue period. We repeated the analysis comparing the Cspk response to the different reward conditions for trials with and without saccades (Figure 3—figure supplement 2C,D). We found that Cspks differentiated between reward conditions in both cases.

Similar to the licking analysis (see 1A), we calculated the PETH of the cells that responded to the cue aligned to the occurrence of a saccade (Figure 3—figure supplement 2E). Here as well the response was very small. We approximated the contribution of saccades to the Cspk response and found that it was negligible and did not differentiate between reward conditions (Figure 3—figure supplement 2E).

We additionally checked if, within the minority of trials with saccades, differences in saccade velocity or direction could underlie the effect on Cspk rate. At the behavioral level, the distributions of saccade velocities and directions were mostly similar between reward conditions (Author response image 1; Joshua et al., 2015). The median of the distribution of saccade velocities was smaller in the large reward condition than the small (saccades tended to be slower in the large reward condition; Signed-rank, p<0.001, n = 9833 saccades). However, this difference cannot solely underlie the coding of reward since reward size is coded even without saccades. Furthermore, this seems to be the opposite of what we would expect if saccade velocity were the underlying cause of the difference in firing rate during the cue. We also analyzed trials with slow and fast saccades separately. There was an increase in Cspk rate following the large reward cue in both cases, although we only used a small fraction of trials for each condition (only 32% of trials overall).

Lastly, we tested whether reward condition interacted with saccade direction, aligned to the preferred direction of the simple spikes of the same cell. We calculated the Cspk rate after the cue in trials with different saccade directions and shifted them so that the PD of simple spikes was to the right (Author response image 1). There was an interaction between reward condition and saccade direction and in particular a large difference between reward conditions at the simple spike null direction (to the left). However, the difference was half a spike, which occurred in a minority of trials and does not explain the effect we observed. We briefly address the velocity and direction analysis in the paper.

**Author response image 1. respfig1:** Analysis of saccade velocity and direction during the cue. (**A**) The distribution of saccade velocity in the 700 ms following the cue and proceeding target motion in both reward conditions. (**B**) The distribution of saccade directions in the same time period as **A.** (**C**) PSTHs aligned to cue for trials with slow and fast saccades (below the 40^th^ and above the 60^th^ percentile respectively). (**D**) The Cspk response following the cue in trials with different saccade directions. The right direction represents the PD of simple spikes for the same cell.

c) A key question is whether there is any cue-related SS modulation – might throw light on the function of the Purkinje cells that were not tuned for visual target direction and on events around the time of the cue related to the complex spikes.

We agree with the reviewer that understanding the relationship between complex and simple spikes is critical. We therefore expanded the analysis and increased the focus on this in the revised manuscript (Figure 6 and Figure 7). We found that the Sspk population exhibited heterogeneous responses. We now note the difference between this heterogeneity and the consistent coding of reward by Cspks. Furthermore, we could not find a link between a cell's simple and complex spike responses to the cue (Figure 7C-D). We address the lack of Cspk-Sspk correlation following the cue in the discussion.

2) Fuller presentation of the results. In general, the presentation of the results was clear, but to fully understand the details, readers must flit between Results section figures and Materials and methods section to try to follow some of the numbers and the specifics of the task.a) Shortly after the cue presentation the data from Figure 2A suggests that the instantaneous discharge rate of complex spikes reaches around 4 Hz. Does this include multiple complex spikes with short Interspike intervals on a single trial? I strongly feel that an illustration of raw data would be very helpful here to allow complex spike and simple spike waveforms to be compared, and that this should have a broader timescale than in Figure 2. For example, this would make a good supplementary data figure.

We added a supplementary figure (Figure 2—figure supplement 1) showing example traces for the response of the cell in Figure 2. We also included histograms of the number of spikes in the 100-300 ms time bin following the cue for cells that responded to the cue presentation (Figure 2C,F).

b. The description of the task in subsection “Complex spikes encode the size of the expected reward” does not fully describe what happens: the coloured target steps in one direction before moving linearly in the direction orthogonal to that direction. This is buried at the end of the paper in the Materials and methods section, and I suspect a very small proportion of the readers will look at that. It really needs to be before or at the time that the task is described.

We added an explanation of this point when first describing our task in the Introduction section.

c) An issue is that the recordings were sampled from a heterogeneous population of Purkinje cells from the flocculus. The Materials and methods section says there were 149 cells, Figure 2 148.

We corrected this mistake, the correct number in both cases is 148.

d) Scatter plots in Figure 2D shows some complex spikes with very low rates in the 100-300 ms interval for both small and large rewards: this could arise if only very few trials were averaged. I cannot see data on minimum numbers of trials in the paper – it should be made available.

We thank the reviewers for noticing that this information was missing. We added it to the Materials and methods section. We also repeated the analysis using more stringent criteria (a minimum of 10 trials per condition) and this did not alter our results.

e) Differentiation is made in Figure 3D between individual Purkinje cells with significant complex spike modulation to the cue and those in which there was not a significant difference. These numbers need to be available in the manuscript – ideally in the first section of the results. At present, it is buried in the Materials and methods section at the end.

We added these numbers to the Results section.

f) Following from this, an example cell is shown in Figure 4 which had both cue-related reward modulation and directional modulation. Again, some numbers on these would be valuable; how many of the sample were significantly modulated by both?

In response to this comment, we performed additional analyses on the subpopulations of cells that responded to cue presentation and cells that were directionally tuned. Different approaches to analysis resulted in opposite results and effect directions. We therefore chose to remove Figure. 7 from the paper, as our data do not clearly show that these are two non-overlapping subpopulations. Instead, we now provide a more nuanced presentation in which we presented examples for the encoding of either or both variables (Figure 4—figure supplement 1 in the revised manuscript) and added the requested numbers to the paper.

g) Figure 6 compares the complex and simple spike modulations, but does not say anything about simple spike modulation related to the reward predicting the cue. At least a brief mention is needed – were any significantly modulated? Is so, were they different for large and small reward predicting cues?

We added a figure (Figure 6 in the revised manuscript) to address this point.

h) The final paragraph of subsection “Representation of reward and target motion in the population” states that the motion tuned cells add an early response which was not selective to reward, which is clear in Figure 7B. Is this a qualitative description or were these significant differences? Since these are population curves, does this reflect heterogeneity in the response pattern, or is it a genuine representation of the behaviour of individual cells?

We do not have enough data to make this claim quantitatively (see our response to comment 2F).

i) How were the cross correlations between CS and SS calculated? (subsection “Data analysis”). One might expect that the authors would calculate the correlations of the two types of spikes triggered on the onset of the cue or pursuit or some other trigger related to the task. It seems that instead they have calculated it based on CS triggered SS PSTHs. This is strange since this is typically the way that people verify that the CS is coming from the same Purkinje cell as the SS. Could it be that in CS with low negative correlations with SS the CS and SS are not actually coming from the same Purkinje cell?

The description of the analysis was not clear enough. We have rephrased it in the revised manuscript. This paragraph referred to Figure 1—figure supplement 2D. The purpose was indeed to verify that there was a pause in Sspk rate following a Cspk, as has previously been shown. We now reference the figure in the Materials and methods section to make this clearer. The correlation now shown in Figure 7C and D was the average responses and not the trial-by-trial activity.

j) The MRI images in Figure 1—figure supplement 2 don't do much to help convince readers that the authors are recording in or near the VPFL. At the very least, some labels would help. Even better would be if the imaging can be done with an electrode or similar placed in a recording track (which would hopefully reside in the VPFL).

We added labels to Figure 1—figure supplement 2A. We cannot perform the MRI scan again, since the chambers were removed from both monkeys. Our main criteria for defining the location was the response of the cells to pursuit eye movements. All other cells were recorded on days when we searched for the eye movement neurons. Our ability to reconstruct the recording sites is very limited, probably due to errors in the exact angle of the guide, and error that is especially enhanced in deep brain recordings. We attempted to calculate the cell's location on the MRI images, but this was not accurate and resulted in some cells being placed outside the brain, or cells with clearly identified complex spikes and simple spikes falling outside the cerebellum.

k) In the Conclusion, the statement that the paper demonstrates "how climbing fibres encode predicting the reward size" is not really appropriate – it shows how a subpopulation of CSs innervating Purkinje cells in the flocculus have activity that encodes predicted reward.

We replaced this statement with a more accurate one.

l) I also disagree with the statement that the authors have "found signals that can be used by the cerebellum to drive behaviour that maximises upcoming reward" – that is entirely possible, but is not shown in this paper – may be used would be more appropriate.

We replaced this statement with a more accurate one.

3) Additional discussion of the potential function of the CS activity related the reward cue.It is not clear what causal role the authors believe the "reward prediction" signal is playing in the task the monkeys are performing. The cue-related CSs come at a time when the Purkinje cells are not engaged in movement and relate differently to ongoing SS compared to CS occurring during movement – in this case what is their function? On the one hand, the authors show that the pursuit velocity is higher on the large reward trials; but, on the other hand, they show that the Purkinje cells with the highest "reward prediction" signals are the least tuned for the direction of pursuit. How do the authors think these untuned Purkinje cells fit within the pursuit control system? Do they even have access to the motor neurons controlling the eye? If not, why does the cerebellum need such a "reward prediction" signal during this task? Could the increased velocities on the large reward trials not just as easily be explained by increased attention/motivation that is not necessarily dependent on the cerebellum?This is especially relevant since the majority of the neurons recorded by the authors did not appear to be within the VPFL, at least as it has been defined by Lisberger and others. That is, the percentage of Purkinje cells with directionally tuned simple spikes is very low compared to what is typically reported for VPFL Purkinje cells and few of the CSs seem to encode retinal slip signals. If the Purkinje cells are not in the VPFL how do the authors know they are even causally involved in the task? More information should be provided about the relative locations of the recorded cells that are tuned and untuned with reference to what is known in the literature about non-visual CS receptive fields in the areas around the flocculus.

We agree with the reviewer that it is extremely important to move beyond representation to a mechanistic understanding of how reward affects cerebellar computation and behavior. We think this is one of the main upcoming challenges of the field. In fact, when we designed the experiment this was exactly what was on our mind. Reward coding at motion onset might have led to linking the behavioral effect, complex spikes and simple spikes, like what was found in the cerebellum during learning. However, this is not what we found. It is possible that the difference in behavior (Figure 1B,C) was not caused by the Cspk signal, so we were careful not to claim that it did and have made this clearer in the discussion in the revised paper. This signal could be involved in the learning of the associations, when the difference in behavior is acquired, but our experiment does not allow us to test this.

We did not manage to accurately calculate the locations of individual cells (see our response to comment 2J), so it is possible that some the neurons we recorded were outside the eye movement areas. We therefore focused on the existence of this signal during the cue and refrained from making claims as to their involvement in movement. When discussing movement-related activity (Figure 4 and Figure 5) we only included cells that were directionally tuned. We reviewed the literature and discussed this with other researchers to find references to non-eye movement areas that surround the Flocculus but could not find information that we could relate to our results.

A weakness of the task design is that it is impossible to distinguish between quantitative versus relative reward size encoding. Although the authors do not claim to study this directly, it is implicit in the learning models cited that there be quantitative reward size encoding (see point #2).

We agree with the reviewer that the coding might be relative rather than absolute. We mention this point in the discussion to prevent over-interpretation of our data.

[Editors' note: further revisions were requested prior to acceptance, as described below.]

Essential revisions:To be fair, this issue is also a potential confound for many or all of the reports that have been coming out about reward coding in the cerebellum. The very exciting thing about the present experimental design is that it has the potential to nail this issue in a way that other studies have not been able to. In contrast to much of the work on reward coding in the cerebellum, which has been done in regions of the cerebellum where very little is known about coding, in this study, the recordings were made from the floccular complex of the cerebellum, where the signals carried by the simple spikes and complex spikes are probably better understood than anywhere in the cerebellum. And what is known about coding by the complex spikes is that they encode image motion on the retina, or retinal slip, in a particular, preferred direction (almost always contraversive or up). Therefore, the most important potential confound for the claim that the complex spikes encode upcoming reward size, is the possibility that the expectation of a large reward causes the animal to make smooth eye movements (slow drift) that results in retinal slip in the preferred direction of the climbing fiber. The authors analyzed licking and saccades, but the most important potential 'lurking variable" to worry about is the known responsiveness of complex spikes in this region of cerebellum to retinal slip at speed of a few degress/s, and that was not convincingly analyzed. The good news is that is quite do-able, for example, by doing a complex spike-triggered average of retinal slip, and comparing the results for the period after cue presentation with those during target motion in the "on" direction for the complex spikes. Unfortunately, the number of cells that were both responsive to the cue and directionally tuned (12 cells) may not be sufficient, hence more cells may need to be recorded.

Several lines of evidence suggest that the response to the reward cue is not a result of the retinal slip caused by drifts of the eye. We used both our data and behavioral data from four additional monkeys recorded using coils on similar tasks to control for the possible confound of drift movements. We present these controls in Figure 4—figure supplement 2 of the revised manuscript.

We recorded eye movements using an eye tracker camera. We follow the lead of some groups that use eye tracker data to analyze drift (Engbert and Kliegl, 2004; Herrmann et al., 2017). However, we note that others show that camera measurement noise might dominate the drift signal (Kimmel et al., 2012; Ko et al., 2016). We therefore first reduced this measurement noise to the best of our ability. The eye tracker measurement of vertical eye velocity was highly correlated to the pupil area measurement we collected simultaneously (Author response image 1). We therefore fitted a linear model between pupil size and eye position for the averages of each recording session. Based on the model, we predicted the measured drift from the pupil size for each trial and subtracted it from the measured drift to obtain an assessment of the drift that is independent of the pupil size artifact. We used these data for the analysis in Figure 4—figure supplement 2A-C and Figure 4—figure supplement 2D second column.

Even after correcting for the pupil artifacts, the signal might still have been dominated by measurement noise. We therefore also used behavioral data recorded on similar tasks using a coil to assess the size of the drift (Author response image 2). In some monkeys the drift differed between reward conditions. However, the direction of the difference was not consistent, and in some monkeys, there was no difference at all. Even in the monkey with the largest difference between drift conditions (Monkey R in Author response image 2) the difference between large and small rewards was very small (~0.2 deg/s). We used these traces to show that the contribution of the drift to the firing rate is expected to be very small (Figure 4—figure supplement 2D).

Lastly and importantly, many of the cells that responded to the cue did not respond during target motion. Therefore, unless these cells responded specifically to small drifts or small retinal slips (which has never been documented in the cerebellum) the reward related response in these cells was not related to the drift.

Overall, the controls above suggest that it is unlikely that the tiny drift at motion onset would drive the robust response after cue onset.

**Author response image 2. respfig2:** Correcting the influence of pupil size on eye position measurements. (**A**) Vertical eye position aligned to cue presentation, for each reward condition. (**B**) Horizontal eye position aligned to cue presentation, for each reward condition. (**C**) Pupil area in arbitrary units aligned to cue presentation, for each reward condition. (**D**) Distribution of R^2^s for models fitting pupil area to vertical (up) or horizontal (down) eye position for each recording session (vertical median = 0.94, horizontal median = 0.17, n = 208).

**Author response image 3. respfig3:** Drift following reward size cue presentation, measured using a coil. (**A-D**) Vertical drift velocity measured using coil following reward cue presentation. (**E-H**) Horizontal drift velocity measured using coil following reward cue presentation.

[Editors' note: further revisions were requested prior to acceptance, as described below.]

The analysis provided in Figure 4—figure supplement 2B is helpful.Figure 4—figure supplement 2A,C would be more helpful if the variance was provided as well as the mean. If the mean was zero and the variance was 10deg/s, there would be a lot more image motion in the preferred direction than if the mean was zero and variance was zero. This matters since there is an asymmetry in the ability of preferred ad anti-preferred direction image motion to increase and suppress the climbing fiber firing rate because of the low basal firing rate and floor at zero.The analysis provided in Figure 4—figure supplement 2D is based on an assumption that is not supported by the literature--that the climbing fiber response to retinal slip increases linearly with the velocity of image motion. On the contrary, in rabbit, cat, and rat, climbing fibers in the flocculus are tuned for low retinal slip speeds, and their responses fall off with speeds >1-2{degree sign}/s (Simpson and Alley, 1974; Blanks and Precht, 1983; Kusunoki et al., 1990; Fushiki et al., 1994). In monkeys, some climbing fibers respond to higher speeds, yet the assumption of a linear increase with image velocity is not supported (Noda et al., 1987; Hoffmann and Distler, 1989; Guo et al., 2014).It is still not clear to me why the authors would not do the obvious thing of a complex spike triggered average of eye velocity in the preceding ~200 ms (for complex spikes in the period associated with the reward cue). After going through the trouble of correcting for pupil size artifacts in the camera data, why not give the climbing fiber spike-triggered eye velocity average a try? Those measurements might be dominated by noise, yet it would still be nice to know that nothing could be detected. Because if something is detected despite the measurement limitations, that would require a reinterpretation of the main resultEssential revisions:Remove Figure 4—figure supplement 2, panel D, unless the authors can provide some justification for the assumption of linear increase in climbing fiber response with image velocity.

In Figure 4—figure supplement 2, the SEMs were included but were small. We replaced them with STDs to make the comparison between variances possible (Figure 4—figure supplement 2A-D in the revised paper). We removed Figure 4—figure supplement 2 as requested. We added the Cspk-triggred drift (Figure 4—figure supplement 2E in the revised paper).

References

Engbert R, Kliegl R. 2004. Microsaccades Keep the Eyes’ Balance During Fixation. Psychol Sci 15:431–431.

Herrmann CJJ, Metzler R, Engbert R. 2017. A self-avoiding walk with neural delays as a model of fixational eye movements. Sci Rep 7:12958.

Ko H, Snodderly DM, Poletti M. 2016. Eye movements between saccades: Measuring ocular drift and tremor. Vision Res 122:93–104.